**Data Availability Statement:** The data that support the findings of this study are held in the data

# Audit of antibiotic utilization patterns and practice for common eye infections at the ambulatory clinic of a teaching hospital in Ghana: Findings and implications

Israel Abebrese Sefah[1]*, Anthony Martin Quagraine[1], Amanj Kurdi[2,3,4,5], Steward Mudenda[6], Brian Godman[3,4]

1 Pharmacy Practice Department, School of Pharmacy, University of Health and Allied Sciences, Volta Region, Ho, Ghana, 2 College of Pharmacy, Al-Kitab University, Kirkuk, Iraq, 3 Strathclyde Institute of Pharmacy and Biomedical Science (SIPBS), University of Strathclyde, Glasgow, United Kingdom, 4 Department of Public Health Pharmacy and Management, School of Pharmacy, Sefako Makgatho Health Sciences University, Ga-Rankuwa, South Africa, 5 College of Pharmacy, Hawler Medical University, Kurdistan Region, Erbil, Iraq, 6 Department of Pharmacy, School of Health Sciences, University of Zambia, Lusaka, Zambia

* isefah@uhas.edu.gh

## Abstract

### Background

Antimicrobial resistance (AMR) is a serious public health issue which is exacerbated by increased inappropriate use of antibiotics for common eye infections. This cross sectional survey was to assess the appropriate use of antibiotics for eye infections in an ambulatory clinic in Ghana and possible determinants.

### Method

The medical records of all patients who sought eye care between January 2022 to December 2022 and were prescribed antibiotics were extracted from the hospital's electronic database. Descriptive, bivariate and multivariate analyses were then conducted.

### Results

A total of 1925 patient medical records were extracted, whose median age was 40 years (IQR 26–69), and were mostly females (58.91%, n = 1134/1925). The eye condition commonly treated with antibiotics was bacteria conjunctivitis (33.51%, n = 645/1925). The most prescribed antibiotic was gentamycin (22.96%, n = 442/1925) followed by ciprofloxacin (16.78%, n = 321/1925). These were mostly topical dosage forms (82.13%, n = 1581/1925). Systemic antibiotics prescribed were mostly from the WHO 'Access' class (83.33%, n = 280/338). The appropriate choice of antibiotic prescribed was 42.44% (n = 817/1925) and this was positivity associated with age (p<0.001), number of antibiotics prescribed (p <0.001), the prescription of topical dosage forms (p <0.001), and WHO 'Access' antibiotic class (p <0.034).

repository of Infectious Disease Data Observatory (https://www.iddo.org/ & https://share.iddo.org/mycontributions).

**Funding:** The author(s) received no specific funding for this work.

**Competing interests:** The authors have declared that no competing interests exist.

## Conclusion

The level of appropriateness of antibiotic prescriptions for eye infections was sub-optimal. Antimicrobial stewardship programs, including prescriber education on guidelines and prescription audit to address associated factors, must now be instigated in this hospital to improve future antibiotic use and prevent the rise of AMR.

## Introduction

Antimicrobial resistance (AMR) is a serious public health issue due to its impact on morbidity, mortality and healthcare cost worldwide [1–3]. AMR could be exacerbated by self-treatment with antibiotics including both oral and topical antibiotics for common eye infections, usually accessible from over-the counter drug stores and pharmacies without a prescription [3, 4]. AMR may also arise from inappropriate prescribing of broad spectrum antibiotics, especially systemic antibiotics, for viral causes and self-limiting forms of the common bacterial eye infections [5, 6]. There have been increasing reports of resistance to some antibiotics used for the management of common bacterial eye infections, which has led to reliance on newer generations of antibiotics further increasing the risk of AMR [7–9].

Damage to the eye due to infections is associated with increased morbidity especially at ambulatory clinics worldwide [10, 11]. Infection and subsequent inflammation of the eye may result in permanent disability such as blindness if prompt and effective therapy, including antibiotics where pertinent, is not administered [12]. The World Health Organization (WHO) estimated that in October 2019 approximately 2.2 billion people worldwide were visually impaired or blind, of which 1 billion cases, which included eye infections, could have been prevented if appropriately treated. 90% of those affected lived in low-and middle income countries (LMIC) like Ghana [13, 14]. Vision impairment is responsible for an appreciable global financial burden, with an estimated annual global productivity loss of US$ 411 billion per year; however, the full cost is likely higher [15]. Some of the treatable or preventable causes of vision loss include cataract, glaucoma, diabetic retinopathy and various forms of eye infections [16].

External and intraocular eye infections are the two main types of eye infections [12]. Conjunctivitis, blepharitis, keratitis, endophthalmitis, dacryocystitis and orbital and periorbital cellulitis are some of the most prevalent eye conditions caused by infections that are commonly diagnosed in ambulatory clinics [17–20]. About 70% of the common eye infections are known to be caused by bacteria leading to increased potential for antibiotic use [6]. One such eye infection is endophthalmitis, which is a serious eye infection requiring urgent effective treatment to prevent permanent vision loss [21, 22]. Bacteria keratitis, an infection of the cornea, is another important eye infection that can lead to vision loss.

Bacteria conjunctivitis, one of the commonly reported ambulatory eye infections, is an inflammation of the conjunctiva mucosa, commonly caused *by Streptococcus pneumoniae*, *Haemophilus influenzae* in children and *Staphylococus aureus* in adults [23]. Whilst most management of eye infections involves administration of high concentrations of antibiotics in ocular tissue with topical dosing or direct injection, reducing the possibility of under-dosing, there have still been reports of treatment failure due to AMR against penicillin and some fluoroquinolones [24, 25], which is an increasing concern.

Antimicrobial Stewardship Programmes (ASPs), including an assessment of guideline compliance via clinical audit with feedback to clinicians, has become a useful strategy to improve future antibiotic utilization across sectors [26–29]. Such audits not only allow an assessment of

the appropriateness of antibiotics prescribed for common infectious diseases, but also to identify the quality of antibiotic use via indicators including percentages of the use of culture and sensitivity and biomarkers to support appropriate diagnosis and guideline compliance [30–33]. Another important quality indicator currently being used to assess appropriate antibiotic prescribing is the proportion of 'Access', 'Watch' and 'Reserve' antibiotics prescribed according to the WHO AWaRe classification system, with a recommendation that at least 60% of antibiotics prescribed are from the 'Access' category [34–36]. This system, developed in 2017, promotes the responsible use of antibiotics as it encourages increased selection of the 'Access' group of antibiotics as first line choice for most common infectious disease [35, 37, 38]. This is due to their effectiveness in managing these infections without increasing the risk for AMR [39].

There are growing concerns regarding increased prevalence of eye infections in Ghana coupled with rising rates of AMR in Ghana, similar to other African countries [40–48]. The National Action Plan to tackle AMR was launched in Ghana in 2017 to help reduce AMR rates. This has several initiatives to ensure appropriate use of antibiotics in humans. Initiatives include training of clinical care providers, developing indicators for monitoring antibiotic use and AMR, and ascertaining antibiotic utilization patterns against current guidelines among other ASP interventions in hospitals [26, 41, 42, 49, 50]. There is currently paucity of evidence on the appropriate use of antibiotics for the commonly reported eye infections among ambulatory clinics in Ghana to guide future initiatives. Consequently, this study was undertaken to address this concern by instigating an audit of the appropriate use of antibiotics in an ambulatory eye clinic in Ghana combined with assessing factors influencing their use.

## Materials and methods

### Study design

A retrospective cross-sectional survey was conducted by extracting the medical records of all patients who sought eye care at the ambulatory clinic of the Ho Teaching Hospital (HTH) from January 2022 to December 2022 using the hospital's electronic database.

### Study site and population

This study was conducted at the eye clinic of HTH, a 306-bed capacity tertiary health facility located in the Volta Regional of Ghana whose capital is Ho., They have a staff strength of 1,200 and 14 wards for patient admission [51]. The hospital was chosen for this initial research as it provides several primary clinical care services including surgical, internal medicine, obstetrics & gynecological, child health and public health, pharmaceutical and diagnostic services for patients in the Region. Consequently, the hospital attracts a large number of patients. The facility also provides several specialized services such as eye, ear, nose and throat (ENT), mental and diabetic cares to patients in the municipality and beyond, and is the only teaching hospital in the Volta Region of Ghana. Consequently, if there are concerns with the prescribing of antibiotics in this setting, these concerns would likely be exacerbated in lower care settings including primary care clinics in this region and others in Ghana.

Overall, HTH provides services to over 20,000 outpatients and inpatients every month. It is the only tertiary hospital in the Volta Region, one of the 16 administrative regions in Ghana, with a 2021 population of 1,659,040 [52]. The eye clinic serves approximately 250 outpatient cases every month. Records of ambulatory patients who have visited the eye clinic within the study period were assessed.

### Inclusion and exclusion criteria

Patients of all ages attending the eye ambulatory clinic to whom antibiotics were prescribed during the study period were included. Patients having eye conditions who were managed with antibiotics on admission were included whereas patients who were not prescribed antibiotics for their eye condition were excluded. Patients with incomplete medical information were also excluded. The medical records of any subsequent visit (s) of a patient with multiple visits within the study period were removed to avoid multiple entry for the same patient.

### Sample size and sampling technique

No sampling was undertaken as all patients who attended the eye ambulatory clinic within the study period were all included in this study. However, an expected annual sample of participants using the Raosoft Inc. online calculator, assuming a 50% appropriateness of antibiotic prescription, an average monthly outpatient attendance of 600, 95% confidence interval, and a 5% margin of error was 1824 patients [53].

### Data collection

Data collection from the hospital's Lightwave Health Information Management System (LHIMS), which is an electronic medical record system were accessed between 1st June 2023 to 31st 154 July 2023. Individual patient data that had patient identifiers were extracted from the LHIMS. An adapted data collection checklist was used, which was similar to the data collection form for audits of ambulatory care management in hospital settings of other LMICs [54, 55]. This approach had been used previously by some of the co-authors where no specific forms exist [26, 31, 56]. The checklist included socio-demographic information including patients' age, gender and national health insurance (NHIS) status (to assess the influence of payment status on prescribers' behavior), clinical information (including the principal eye infection diagnosed), names of antibiotics prescribed and their AWaRe classification, whether a culture and sensitivity test was requested (to assess the proportion of targeted treatments among the eye infections diagnosed), and number of antibiotics prescribed per patient encounter. Appropriateness was determined using the Seventh Edition of the Ghana STG for management of ENT infections in terms of the choice [57]. When the condition was not discussed in the Ghanaian STG, the WHO AWaRe antibiotic book was used as a reference point [39] (Table 1).

## Data handling

The collated data was coded to safeguard patient identity by removing personal identifiers and password protected to the principal investigator.

### Data analysis

Data extracted onto the checklist was transferred into a Microsoft Excel sheet before being exported to STATA version 14 (StrataCorp, College Station, TX, USA) for analysis. The appropriateness of prescribed antibiotics based on the choice for the diagnosed eye conditions was the primary study outcome measure. Descriptive statistics were used to determine the median age and the proportions of each categorical variables and Chi square test of independence was performed to assess association between the outcome variable and the various independent variables. A logistic regression analysis was also performed using all statistically significant independent variables from the bivariate analysis (p value <0.05 at 95% confidence interval).

**Table 1. Summary of the common types of bacteria eye infections, choice antibiotics and it's WHO AWaRE class.**

| Bacteria Eye Infection | Causes and clinical presentation | Causative organism | Antibiotic treatment | AWaRE Class of Antibiotic |
|---|---|---|---|---|
| Bacteria Conjunctivitis [38, 57] | Red, watery and itchy eye, feeling of 'sand in the eye, which may be swollen, painful and discharging | Most cases are viral. Common bacteria pathogens are *Chlamydia trachomatis* (serovars D to K) and *Neisseria gonorrhoeae* | STG: Tetracycline 1% eye ointment PLUS Chloramphenicol 0.5% eye drops OR Ciprofloxacin 0.3 eye drops<br>WHO Antibiotic Book: Gentamicin 0.3% eye drops OR Ofloxacin.0.3% eye drops OR Tetracycline 1% eye ointment<br>Use Ceftriaxone 250mg IM single dose OR Azithromycin 1g for gonococcal conjunctivitis | Tetracycline (Access) Chloramphenicol (Access) Gentamicin (Access) Ceftriaxone (Watch) Azithromycin (Watch) |
| Keratitis [38] | Usually painful red eye, decreased vision, more tears and corneal oedema with a feeling of "having something in the eye" and difficulty in keeping the eye open +/- eye discharge | Pseudomonas spp, *Staphylococcus epidermidis, Staphylococcus aureus, Streptococcus pneumoniae* | Ofloxacin 0.3% EYE DROPS 1 drop in the affected eye q1h for 48 hours, then q4h until healed.<br>NB: Drops are preferred over ointment due to better corneal tissue penetration | Ofloxacin (Watch) |
| Endophthalmitis [38] | Usually painful red eye, blurred vision and trouble looking at bright light | Coagulase-negative Staphylococcus, *Staphylococcus aureus,* Streptococcus spp., Klebsiella spp., Bacillus cereus | WHO Antibiotic Book: Vancomycin 1mg PLUS Ceftazidime 2.25mg (intravitreal injection)<br>If endogenous infection is present, add Ceftriaxone 2g IV<br>+ Vancomycin 15-20mg/kg 12h IV | Ceftriaxone (Watch) Ceftazidime (Watch) Vancomycin (Watch) |
| Periorbital Cellulitis [38] | Unilateral signs of red, swollen, warm and tender eyelid with no restricted or painful eye movement +/- fever (≥ 38.0˚C) | *Staphylococcus aureus, Streptococcus pneumoniae, Haemophilus influenzae, Moraxella catarrhalis,* Anaerobes | WHO Antibiotic Book: Amoxicillin + clavulanic acid 625mg Oral OR 1.2g IV OR<br>Cefalexin 500mg oral OR Cloxacillin 500mg oral OR 2g IV | Amoxicillin + clavulanic acid (Access) Cefalexin (Access) Cloxacillin (Access) |

## Ethical consideration

Ethical approvals were secured from both the Research Ethics Committees of the University of Health and Allied Sciences (UHAS-REC A.7 [53] 22–23) and the Ho Teaching Hospital (HTH-REC (37) FC_2022. Individual patient identifiers anonymized during the data collection to safeguard confidentiality. Patient informed consent were not collected as there were no direct contact with the as data were obtained solely from their medical records.

## Results

### Patient characteristics

A total of 1925 medical records of patients who visited the HTH and were prescribed with antibiotics for a diagnosed eye infection were retrieved from the LHIMS database. The median age of the patients was 40 years (IQR 26–69), with most (51.95%, n = 1000/1925) patients between aged of 25 to 64 years, followed by those above 65 years (27.27% (n = 525/1925). Most of the patients were females (58.91%, n = 1134/1925). The majority (91.32%, n = 1758) of the patients had registered and were active members of the Ghana National Health Insurance (NHIS). The commonest eye condition that was treated with antibiotics was bacteria conjunctivitis (33.51%, n = 645/1925), followed by those with allergic conjunctivitis (17.75%, n = 337/1925), with 9.30% (n = 179/1925) of patients that were given antibiotics having no documented eye or any other infectious disease. 99.38% (n = 1913/1925) of the patients had no additional diagnosis, with dry eye syndrome (0.16%, n = 3/1925) being the highest documented additional diagnosis when this occurred. The most prescribed antibiotic was gentamycin eye drops

(22.96%, n = 442/1925) followed by ciprofloxacin drops (16.78%, n = 321/1925) and chloramphenicol drops (10.03%, n = 193/1925). Most of the patients received only the topical form of antibiotics (82.13%, n = 1581/1925), followed by those who received both topical and oral antibiotics (9.40%, n = 181/1925), while 8.36% (n = 161/1925) received only the oral form. The most common oral antibiotic prescribed was amoxicillin (50.34%, n = 145/288) followed by co-amoxiclav (17.71%, n = 45/288). Aminoglycosides (25.2%, n = 487/1925) were the majority class of antibiotic that was prescribed followed by those in the quinolone class (16.6%, n = 321/1925). A greater proportion of the systemic antibiotics prescribed were from the 'Access' group (83.33%, n = 280/338) of the WHO AWaRe classification while 13.99% (n = 47/338) were from 'Watch' group. Only 42.44% (n = 817/1925) of the antibiotics prescribed was appropriate based on compliance with the choice of antibiotics in the Ghana STG/ AWaRe book (Table 2).

### Bivariate analysis

The Chi square test of independence showed a positive significant association between the appropriateness level and younger age groups of patients (p<0.001), prescription with fewer number of antibiotics (p <0.001), the use of topical dosage forms of antibiotics (p <0.001), and the prescription of WHO 'Access' group of the antibiotics (p <0.034). The gender status, NHIS status, and the principle diagnosis documented showed no significant association with the outcome indicator (Table 3).

### Multiple logistic regression

A multiple logistic regression between appropriateness based on choice and all the variables that had significant bivariate association showed that antibiotic appropriateness was only independently predicted by the age group of the patients (aOR = 0.62, CI = 0.43–0.88, p-value = 0.007) (Table 4).

## Discussion

We believe this is the first clinical audit of antibiotic use among patients with different eye infections attending an ambulatory clinic in a leading hospital in Ghana. This builds on the study of Hope et al (2022) that just assessed the management of acute conjunctivitis in a specialist eye hospital in Ghana against current treatment guidelines [58]. We also believe such an audit of a particular specialty is useful as it provides evidence on the commonly reported infectious disorders within that specialty, the types of antibiotics prescribed, their appropriateness and factors influencing their prescription. The quality gaps of antibiotic use identified are useful in designing ASPs to improve their future use, achieve improved patient outcomes, and help reduce the negative consequences of rising AMR rate [29, 42]. This is similar to previous studies in Ghana, which assessed the appropriate management of common infectious disease in ambulatory care including respiratory tract infection such as pneumonia, dental infections and urinary tract infections [26, 31, 59–62].

Of the 1925 medical records of patients that were extracted from the hospital's electronic database, patients treated with antibiotic for common eye conditions within the age group of 25 to 50 years were the majority with a median age of 40 years. The most predominant eye infection diagnosed among our study participants was bacterial conjunctivitis, which has been reported to be commonly diagnosed in children than in adults making this inconsistent with our finding [22, 63]. In the young age group, this diagnosis is most prevalent and it is usually associated with epidemics resolving usually within 7 days [64]. Our finding though is

**Table 2. Socio-demographic and clinical characteristics of patients *(n = 1925)*.**

| Variables | Frequency (*n*) | Percentage (%) |
|---|---|---|
| **Median age** (years) (IQR) | 40 (26–69) | |
| **Age (years)** | | |
| 0–14 | 94 | 4.88 |
| 15–24 | 306 | 15.90 |
| 25–64 | 1000 | 51.95 |
| ≥65 | 525 | 27.27 |
| **Gender** | | |
| Male | 791 | 41.09 |
| Female | 1134 | 58.91 |
| **NHIS Status** | | |
| Yes | 1758 | 91.32 |
| No | 167 | 8.68 |
| **Principal Diagnosis** | | |
| Conjunctivitis (bacterial) | 645 | 33.51 |
| Conjunctivitis (allergic) | 337 | 17.51 |
| Ocular pain | 115 | 5.97 |
| Conjunctivitis (viral) | 48 | 4.05 |
| Hordeolum (Stye) | 73 | 3.79 |
| Pterygium | 55 | 2.86 |
| Cataract | 49 | 2.55 |
| Refractive disorders | 47 | 2.44 |
| Cellulitis (orbital) | 40 | 2.08 |
| Uveitis | 32 | 1.66 |
| No documented diagnosis | 179 | 9.30 |
| Others | 305 | 15.90 |
| **Additional diagnosis** | | |
| Dry eye syndrome | 3 | 0.16 |
| Others | 9 | 0.46 |
| No additional diagnosis | 1913 | 99.38 |
| **Overall Antibiotics Prescribed** | | |
| Amoxicillin | 62 | 3.22 |
| Chloramphenicol | 193 | 10.03 |
| Ciprofloxacin | 323 | 16.78 |
| Gentamycin | 442 | 22.96 |
| Gentamycin+Chloramphenicol | 257 | 13.35 |
| Gentamycin+Tetracycline | 76 | 3.95 |
| Tetracycline | 154 | 8.00 |
| Tetracycline+Ciprofloxacin | 50 | 2.59 |
| Other | 368 | 19.11 |
| **Number of Antibiotics** | | |
| One | 1305 | 67.79 |
| Two | 589 | 30.60 |
| Three | 30 | 1.56 |
| Four | 1 | 0.05 |
| **Dosage form** | | |
| Oral | 161 | 8.36 |
| Oral &Topical | 181 | 9.40 |

*(Continued)*

**Table 2.** (Continued)

| Variables | Frequency (n) | Percentage (%) |
|---|---|---|
| Topical | 1581 | 82.13 |
| Topical & Intravenous | 1 | 0.05 |
| Intravenous | 1 | 0.05 |
| **WHO Aware Grouping for Systemic Antibiotics (n = 336)** | | |
| Access | 280 | 83.33 |
| Watch | 47 | 13.99 |
| Access + Watch | 9 | 2.68 |
| **Topical antibiotics prescribed (n = 1582)** | | |
| Gentamycin | 431 | 27.24 |
| Ciprofloxacin | 307 | 19.41 |
| Gentamicin + Chloramphenicol | 254 | 16.06 |
| Chloramphenicol | 190 | 12.01 |
| Tetracycline | 148 | 9.36 |
| Gentamycin + Tetracycline | 74 | 4.68 |
| Others | 178 | 11.25 |
| **Oral antibiotics prescribed (n = 288)** | | |
| Amoxicillin | 145 | 50.34 |
| Co-amoxiclav | 51 | 17.71 |
| Azithromycin | 22 | 7.63 |
| Ciprofloxacin | 10 | 3.47 |
| Others | 60 | 20.83 |
| **Appropriate choice of antibiotic use (n = 1925)** | | |
| Yes | 817 | 42.44 |

NB: Others in additional diagnosis included Acute sinusitis, acute pain, acute upper respiratory infections, allergic rhinitis, pelvic inflammatory disease, pterygium and tension-type headache while others in the principal diagnosis included blindness, cataract, endolphthalmitis, diabetic retinopathy, chalazion, blepharitis, conjunctival hemorrhage, dacryocystitis, episcleritis, eye injuries, optic nerve disorders, staphyloma, foreign body in eye, corneal ulcer, glaucoma, and keratitis

consistent with a similar audit conducted in the eye clinic of a tertiary hospital in Ethiopia and primary care facility in Ghana [21, 43].

Other common eye disorders seen in our study that were treated with antibiotics included hordeolum, orbital cellulitis, and pterygium. This is consistent with other studies that have assessed commonly reported eye disorders that are directly caused by bacterial infections [17, 18, 44, 45, 58]. There was however misuse of antibiotics for some of the conditions in our study including allergic conjunctivitis, ocular pain, and viral conjunctivitis. This is a concern as these conditions are not known to be directly caused by bacteria; consequently, making the prescribing of antibiotics inappropriate. Moreover, without the support of culture and sensitivity, most inflammatory eye conditions may have been assumed to have bacterial secondary infections leading to the irrational and overuse of antibiotics [43, 63, 65].

Overall, there was suboptimal appropriateness of the use of antibiotics in our study with only 42.44% of antibiotics used considered appropriate according to the Ghana Standard Treatment Guidelines or AWaRe antibiotic guidelines [39, 57]. This is different compared to another study undertaken in Ghana which recorded over 70% appropriate use of antibiotic for acute conjunctivitis in an ambulatory care setting [58] though this audit was undertaken in a specialist eye hospital. The low level of appropriateness of antibiotics in our study may have resulted from prescribers' irrational antibiotic use for diagnosed conditions including viral conjunctivitis, allergic conjunctivitis, ocular pain, and other inflammatory eye diseases with no

**Table 3. Association between socio-demographic and clinical characteristics and appropriateness of antibiotic choice based on the Ghana Standard Treatment Guidelines (STG).**

| Variables | Total, n (%) | Appropriateness based on choice | | p-value |
| --- | --- | --- | --- | --- |
| | | Inappropriate | Appropriate | |
| **Gender (n = 1925)** | | | | 0.592 |
| Male | 791 (41.09) | 461(58.28) | 330 (41.72) | |
| Female | 1134 (58.91) | 647(57.05) | 487 (42.95) | |
| **Age group (n = 1925)** | | | | **<0.001** |
| 0–14 | 94 (4.88) | 18 (19.15) | 76 (80.85) | |
| 15–24 | 306 (15.90) | 172 (56.20) | 134 (43,79) | |
| 25–64 | 1000 (51.95) | 588 (58.80) | 412 (41.20) | |
| ≥65 | 525 (27.27) | 330 (62.86) | 195 (37.14) | |
| **NHIS Status (n = 1925)** | | | | 0.146 |
| Insured | 1758 (91.32) | 1003 (57.05) | 755(42.95) | |
| Non-insured | 167 (8.68) | 105(62.87) | 62 (37.13) | |
| **Number of antibiotics (n = 1925)** | | | | **<0.001** |
| One | 1305(67.79) | 865(66.28) | 440(33.72) | |
| Two | 589(30.60) | 221(37.52) | 368(62.47) | |
| Three | 30(1.56) | 21(70) | 9 (30) | |
| Four | 1(0.05) | 1(100) | 0 (0) | |
| **Dosage Form (n = 1925)** | | | | **<0.001** |
| Intravenous | 1(0.05) | 1(100) | 0 (0) | |
| Oral | 161(8.36) | 115(71.43) | 46(28.57) | |
| Oral + Topical | 181(9.40) | 123(67.96) | 58(32.04) | |
| Topical | 1581(82.13) | 868 (54.90) | 713(45.10) | |
| Topical + Intravenous | 1(0.05) | 1(100) | 0(0) | |
| **WHO Aware Category (n = 388)** | | | | **0.034** |
| Access | 280 (83.33) | 187 (66.79) | 93 (33.21) | |
| Access + Watch | 9 (2.68) | 9 (100.00) | 0 (0.00) | |
| Watch | 47 (13.99) | 37 (78.72) | 10 (21.28) | |

**NB**: Emboldened p-value are those that are below the significance level of 0.05.

proven bacterial infection, which should have been managed symptomatically with topical analgesics, steroid, anti-inflammatory and anti-allergic treatments [66, 67]. This situation may also have been worsened by the use of misuse of antibiotics in 9.30% of the patients with no documented eye infection diagnosis (Table 1). This contrasts with the study of Hope et al (2022), which reported appreciable appropriateness of antibiotic prescribing in patients with acute conjunctivitis which was commonly diagnosed among children (58). Though appropriateness was only predicted by patients' age, it was associated with the use of more "Access' group antibiotics over 'Watch' group, increased use of topical antibiotics compared to the oral dosage forms and the prescription of fewer combinations of antibiotics for diagnosed eye infections. These quality indicators in the guidelines must be emphasized during education of clinicians in this hospital as part of a broader ASP to curb rising AMR rate. Similar suboptimal use of appropriate choice of antibiotics has been observed in respiratory tract infections especially pneumonia and dental infections in Ghana (26, 31). Such misuse and overuse of antibiotics seen in our study have the potential to trigger genetic mutations through selective pressure on non-resistant bacteria driving high AMR rate [40].

**Table 4. Multiple logistic regression between clinical characteristics and antibiotic choice appropriateness based on the Standard Treatment Guidelines.**

| Variables | Odds Ratio | Confidence interval (95%) | p-value |
|---|---|---|---|
| **Age** | 0.62 | 0.43–0.88 | **0.007** |
| – 14 (r) | 1 | | |
| 15–24 | 0.13 | 0.01–1.29 | |
| 25–64 | 0.15 | 0.02–1.34 | |
| 65 years and above | 0.07 | 0.01–0.64 | |
| **Number of Antibiotics** | 0.79 | 0.40–1.55 | 0.495 |
| One (r) | 1 | | |
| Two | 0.82 | 0.26–2.61 | |
| Three | 0.73 | 0.17–3.11 | |
| Four | 1 | | |
| **WHO Aware Category (n = 388)** | 0.69 | 0.47–1.02 | 0.060 |
| Access (r) | 1 | | |
| Access + Watch | 1 | | |
| Watch | 0.55 | 0.25–1.21 | |
| **Dosage form prescribed** | 1.32 | 0.58–2.98 | 0.508 |
| Topical (r) | 1 | | |
| Oral | 0.73 | 0.23–2.29 | |
| Oral + Topical | 1 | | |
| Topical + Intravenous | 1 | | |
| Intravenous | 1 | | |

NB: R in bracket indicate the referenced co-variate for the analysis

The commonest antibiotics used for the management of the various diagnosed eye disorders in our study included gentamycin, ciprofloxacin and chloramphenicol. These broad spectrum antibiotics have been recommended for most of the common eye infections diagnosed including bacteria conjunctivitis, bacteria keratitis, orbital cellulitis and infectious endophthalmitis (Table 1) [39, 57]. This is because of their broad coverage over the identified causative pathogens in these eye infections, which are commonly a mixture of gram positive and negative bacteria requiring broad spectrum antibiotics [63, 68]. Empirical treatment of eye infections with antibiotics is recommended when there is sufficient clinical suspicion of bacteria as the causative agent to avoid complications [20, 63]. Bacteria keratitis, for example, is a threatening eye infection which may progress rapidly to corneal destruction within 24 to 48 hours if effective treatment, such as topical fluoroquinolones, is not immediately initiated. However once a pathogen has been isolated, treatment can be modified making it a targeted treatment [25, 69–71]. The antibiotics prescribed in our study were predominantly administered as a topical formulation followed by those with a combination of topical and oral dosage forms. The use of the topical formulations alone is generally effective as they ensure high tissue concentrations with minimal systemic adverse effects. In circumstances where topical dosage forms are not available, topical formulations are sometimes fortified with an additional antibiotic by a pharmacist to eliminate the need for the addition of oral dosage forms [65]. For instance, the recommended treatment of bacteria keratitis is the fortification aminoglycoside with second generation cephalosporin [65]. However, the limited availability, cost and inconvenience of these fortified preparations have led to the increased use of topical fluoroquinolones for bacteria keratitis [64].

Our study also showed that the level of appropriateness reduced with increasing patient's age (p<0.001), the use of more than two antibiotics (p <0.001), the prescription of non-topical dosage forms of antibiotics (p <0.001), and the increased use of oral antibiotics from the WHO 'Watch' group (p <0.034) (Table 4). The Ghana STG recommends the use of a combination of topical tetracycline and chloramphenicol administered separately or topical ciprofloxacin alone for the management of bacteria conjunctivitis; consequently, making the addition of oral antibiotics or topical gentamycin inappropriate [57]. Such practices contributed to the high level of non-compliance with treatment guidelines seen in our study. It was however commendable to see that a greater proportion of systemic antibiotics prescribed were in the WHO 'Access' group as these have a lower risk of causing AMR compared to those in the 'Watch' group [72, 73].

The patient's age was the only predictor of the level of appropriateness of antibiotic prescription for eye infections where adult participants had reduced level of appropriateness compared to their younger counterpart. This could be due to the fact that the adult age group had a higher proportion of eye infections in our study coupled with their increased risk of outbreak of conjunctivitis especially due to viral causes, their increased risk of eye trauma from accidents and natural changes to the eye structures due to aging [23, 74]. Consequently, it makes this age group more prone to common eye disorders which are likely to be seen in ambulatory clinics and be possibly mistreated with antibiotics.

The low level of antibiotics appropriateness observed in this study for one of the commonly reported ambulatory diseases in this leading tertiary hospital is a concern requiring urgent ASP interventions. Documented appropriate interventions include, but not limited to, clinician education of treatment guidelines, prospective audit with feedback to clinicians, clinical meetings to discuss appropriate treatment regimens of common eye infections, and restrictions on the use of oral antibiotics from the 'Watch' group for common eye infections [29, 75].

We are aware of a number of limitations with this study. Firstly, this study is limited by the use of a single study site affecting the generalizability of the findings. Secondary, the use of limited patients' characteristics to assess the determinants of appropriate prescription of antibiotics will introduce possible biases and confounders that might not be accounted for. In spite of these limitations, the use of large sample size and the choice of the study site provides key stakeholders in this hospital with essential baseline data on the level and potential causes of inappropriate use of antibiotics for common eye diseases to guide future quality improvement initiatives as part of effective ASPs.

## Conclusion

There was a low level of appropriate use of antibiotics for diagnosed eye diseases at the ambulatory eye clinic of this teaching hospital, which was associated with patient's age, the use of more than two antibiotics, the prescription of non-topical dosage forms of antibiotics, and the increased use of antibiotics from the 'Watch' group. Appropriateness of the choice of antibiotic was predicted only by patients' age. There must be continued efforts to institutionalize ASPs in the ambulatory care settings in this hospital to improve future management and reduce the rate of AMR.

## Author Contributions

**Conceptualization:** Israel Abebrese Sefah, Anthony Martin Quagraine.

**Data curation:** Israel Abebrese Sefah, Anthony Martin Quagraine, Amanj Kurdi, Steward Mudenda, Brian Godman.

**Formal analysis:** Israel Abebrese Sefah, Anthony Martin Quagraine, Amanj Kurdi, Steward Mudenda, Brian Godman.

**Investigation:** Israel Abebrese Sefah, Anthony Martin Quagraine.

**Methodology:** Israel Abebrese Sefah, Anthony Martin Quagraine.

**Project administration:** Israel Abebrese Sefah.

**Resources:** Israel Abebrese Sefah.

**Software:** Israel Abebrese Sefah, Anthony Martin Quagraine.

**Supervision:** Israel Abebrese Sefah.

**Validation:** Israel Abebrese Sefah, Anthony Martin Quagraine, Amanj Kurdi, Steward Mudenda, Brian Godman.

**Visualization:** Israel Abebrese Sefah, Anthony Martin Quagraine, Amanj Kurdi, Steward Mudenda, Brian Godman.

**Writing – original draft:** Israel Abebrese Sefah, Anthony Martin Quagraine, Amanj Kurdi, Steward Mudenda, Brian Godman.

**Writing – review & editing:** Israel Abebrese Sefah, Anthony Martin Quagraine, Amanj Kurdi, Steward Mudenda, Brian Godman.

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
