## [Decision Letter · Decision Letter 0]

26 Aug 2024

PONE-D-24-26286Audit of Antibiotic utilization patterns and practice for common eye infections at the ambulatory clinic of a Teaching Hospital in Ghana: Findings and implicationsPLOS ONE

Dear Dr. Sefah,

Thank you for submitting your manuscript to PLOS ONE. After careful consideration, we feel that it has merit but does not fully meet PLOS ONE’s publication criteria as it currently stands. Therefore, we invite you to submit a revised version of the manuscript that addresses the points raised during the review process.

Please submit your revised manuscript by Oct 10 2024 11:59PM. If you will need more time than this to complete your revisions, please reply to this message or contact the journal office at plosone@plos.org. Please include the following items when submitting your revised manuscript:A rebuttal letter that responds to each point raised by the academic editor and reviewer(s). You should upload this letter as a separate file labeled 'Response to Reviewers'.A marked-up copy of your manuscript that highlights changes made to the original version. You should upload this as a separate file labeled 'Revised Manuscript with Track Changes'.An unmarked version of your revised paper without tracked changes. You should upload this as a separate file labeled 'Manuscript'.If applicable, we recommend that you deposit your laboratory protocols in protocols.io to enhance the reproducibility of your results. Protocols.io assigns your protocol its own identifier (DOI) so that it can be cited independently in the future. For instructions see: https://journals.plos.org/plosone/s/submission-guidelines#loc-laboratory-protocols. Additionally, PLOS ONE offers an option for publishing peer-reviewed Lab Protocol articles, which describe protocols hosted on protocols.io. Read more information on sharing protocols at https://plos.org/protocols?utm_medium=editorial-email&utm_source=authorletters&utm_campaign=protocols.

We look forward to receiving your revised manuscript.

Kind regards,

Kwame Kumi Asare, Ph.D

Academic Editor

PLOS ONE

Journal Requirements:

2. In this instance it seems there may be acceptable restrictions in place that prevent the public sharing of your minimal data. However, in line with our goal of ensuring long-term data availability to all interested researchers, PLOS’ Data Policy states that authors cannot be the sole named individuals responsible for ensuring data access (http://journals.plos.org/plosone/s/data-availability#loc-acceptable-data-sharing-methods).

3. In the online submission form, you indicated that [The data that support the findings of this study are available and can be accessed from the corresponding author, [IAS], upon reasonable request.]. 

Reviewers' comments:

Reviewer's Responses to Questions

**Comments to the Author**

1. Is the manuscript technically sound, and do the data support the conclusions?

Reviewer #1: Yes

Reviewer #2: Yes

2. Has the statistical analysis been performed appropriately and rigorously? 

Reviewer #1: Yes

Reviewer #2: I Don't Know

3. Have the authors made all data underlying the findings in their manuscript fully available?

Reviewer #1: Yes

Reviewer #2: Yes

4. Is the manuscript presented in an intelligible fashion and written in standard English?

Reviewer #1: Yes

Reviewer #2: Yes

5. Review Comments to the Author

Reviewer #1: This study aims to investigate the level of appropriate use of antibiothics for diagnosed eye diseases at the ambulatory eye clinic of a teaching hospital. The study was conducted in good practice, supported by data and proper statistical analysis. The result showed that level of appripriateness is only 42% (!) and appropriateness of the choice of antibiotic was predicted only by patients’ age. This is quite shocked that author need to emphasize it and provide deeper analysis about this finding and its impact to the patients. It is also suggested to compare to other disease such as pneumonia, dental infections and urinary tract infections for broader perspective of antibiotics management. Please elaborate more in discussion section.

Reviewer #2: Review on topical antibiotics is limitted. This data is important, eventhough this is a descriptive study, however the information is quite important. Some correlations somehow need to be clarify and confirmation such as how to explain the correlation between age with the prescription of antibiotics.

6. PLOS authors have the option to publish the peer review history of their article (what does this mean?). If published, this will include your full peer review and any attached files.

Reviewer #1: **Yes: **Fahmi Fahmi

Reviewer #2: **Yes: **Erni J.Nelwan

---

## [Author Response · Author response to Decision Letter 0]

10 Sep 2024

AUTHORS’ RESPONSE: We have now updated the data availability statement to read "The data that support the findings of this study are held in the data repository of Infectious Disease Data Observatory (https://www.iddo.org/ & https://share.iddo.org/mycontributions).

---

## [Decision Letter · Decision Letter 1]

17 Oct 2024

Audit of Antibiotic utilization patterns and practice for common eye infections at the ambulatory clinic of a Teaching Hospital in Ghana: Findings and implications

PONE-D-24-26286R1

Dear Dr. Sefah,

We’re pleased to inform you that your manuscript has been judged scientifically suitable for publication and will be formally accepted for publication once it meets all outstanding technical requirements.

Kind regards,

Kwame Kumi Asare, Ph.D

Academic Editor

PLOS ONE

Additional Editor Comments (optional):

Reviewers' comments:

Reviewer's Responses to Questions

**Comments to the Author**

1. If the authors have adequately addressed your comments raised in a previous round of review and you feel that this manuscript is now acceptable for publication, you may indicate that here to bypass the “Comments to the Author” section, enter your conflict of interest statement in the “Confidential to Editor” section, and submit your "Accept" recommendation.

Reviewer #1: All comments have been addressed

2. Is the manuscript technically sound, and do the data support the conclusions?

Reviewer #1: Yes

3. Has the statistical analysis been performed appropriately and rigorously? 

Reviewer #1: Yes

4. Have the authors made all data underlying the findings in their manuscript fully available?

Reviewer #1: Yes

5. Is the manuscript presented in an intelligible fashion and written in standard English?

Reviewer #1: Yes

6. Review Comments to the Author

Reviewer #1: This study aims to investigate the level of appropriate use of antibiothics for diagnosed eye diseases at the ambulatory eye clinic of a teaching hospital. The study was conducted in good practice, supported by data and proper statistical analysis. The result showed that level of appripriateness is only 42% (!) and appropriateness of the choice of antibiotic was predicted only by patients’ age. This is quite shocked that author need to emphasize it and provide deeper analysis about this finding and its impact to the patients. It is also suggested to compare to other disease such as pneumonia, dental infections and urinary tract infections for broader perspective of antibiotics management.

7. PLOS authors have the option to publish the peer review history of their article (what does this mean?). If published, this will include your full peer review and any attached files.

Reviewer #1: **Yes: **Fahmi Fahmi

---

## [Editor Report · Acceptance letter]

20 Oct 2024

PONE-D-24-26286R1 

PLOS ONE

Dear Dr. Sefah, 

I'm pleased to inform you that your manuscript has been deemed suitable for publication in PLOS ONE. Congratulations! Your manuscript is now being handed over to our production team.

Kind regards, 

on behalf of

Dr. Kwame Kumi Asare 

Academic Editor

PLOS ONE